# MLe-KCNQ2: An Artificial Intelligence Model for the Prognosis of Missense *KCNQ2* Gene Variants

**DOI:** 10.3390/ijms25052910

**Published:** 2024-03-02

**Authors:** Alba Saez-Matia, Markel G. Ibarluzea, Sara M-Alicante, Arantza Muguruza-Montero, Eider Nuñez, Rafael Ramis, Oscar R. Ballesteros, Diego Lasa-Goicuria, Carmen Fons, Mónica Gallego, Oscar Casis, Aritz Leonardo, Aitor Bergara, Alvaro Villarroel

**Affiliations:** 1Instituto Biofisika, CSIC-UPV/EHU, 48940 Leioa, Spain; alba.saezmatia@gmail.com (A.S.-M.); saraalicante@hotmail.com (S.M.-A.); arantza.muguruza.montero@gmail.com (A.M.-M.); enviadero@gmail.com (E.N.); 2Physics Department, Universidad del País Vasco, UPV/EHU, 48940 Leioa, Spainrafaelramis21@gmail.com (R.R.); orodriguez049@ikasle.ehu.eus (O.R.B.);; 3Donostia International Physics Center, 20018 Donostia, Spain; diego.lasa@dipc.org; 4Centro de Física de Materiales CFM, CSIC-UPV/EHU, 20018 Donostia, Spain; 5Pediatric Neurology Department, Sant Joan de Déu Hospital, Institut de Recerca Sant Joan de Déu, Barcelona University, 08950 Barcelona, Spain; carmen.fons@sjd.es; 6Departamento de Fisiología, Universidad del País Vasco, UPV/EHU, 01006 Vitoria-Gasteiz, Spain; monica.gallego@ehu.eus (M.G.); oscar.casis@ehu.eus (O.C.)

**Keywords:** epilepsy, neurodevelopmental disorder, KCNQ, prognosis, machine learning

## Abstract

Despite the increasing availability of genomic data and enhanced data analysis procedures, predicting the severity of associated diseases remains elusive in the absence of clinical descriptors. To address this challenge, we have focused on the K_V_7.2 voltage-gated potassium channel gene (*KCNQ2*), known for its link to developmental delays and various epilepsies, including self-limited benign familial neonatal epilepsy and epileptic encephalopathy. Genome-wide tools often exhibit a tendency to overestimate deleterious mutations, frequently overlooking tolerated variants, and lack the capacity to discriminate variant severity. This study introduces a novel approach by evaluating multiple machine learning (ML) protocols and descriptors. The combination of genomic information with a novel Variant Frequency Index (VFI) builds a robust foundation for constructing reliable gene-specific ML models. The ensemble model, MLe-KCNQ2, formed through logistic regression, support vector machine, random forest and gradient boosting algorithms, achieves specificity and sensitivity values surpassing 0.95 (AUC-ROC > 0.98). The ensemble MLe-KCNQ2 model also categorizes pathogenic mutations as benign or severe, with an area under the receiver operating characteristic curve (AUC-ROC) above 0.67. This study not only presents a transferable methodology for accurately classifying *KCNQ2* missense variants, but also provides valuable insights for clinical counseling and aids in the determination of variant severity. The research context emphasizes the necessity of precise variant classification, especially for genes like *KCNQ2*, contributing to the broader understanding of gene-specific challenges in the field of genomic research. The MLe-KCNQ2 model stands as a promising tool for enhancing clinical decision making and prognosis in the realm of *KCNQ2*-related pathologies.

## 1. Introduction

Either inherited or *de novo* mutations in the *KCNQ2* gene can result in highly variable phenotypes ranging from healthy normal to severe infantile epileptic encephalopathy [1]. Individualized therapies and genetic counseling are needed in these patients and families and, therefore, early and correct identification of the potential phenotypic severity of the variant is crucial. Rapid advances in genomic technologies have made research and clinical sequencing more accessible [2,3], but identifying true phenotypically causal variants and assessing the prognosis remain challenging [4]. This challenge is particularly pronounced for rare diseases and complex pathologies where genetic information alone is often insufficient [4]. 

This study focuses on pathologies related to the *KCNQ2* gene, which encodes the voltage-gated potassium channel subunit K_V_7.2. Initially linked to self-limited neonatal epilepsy (SeLNE) [5,6], *de novo* missense *KCNQ2* mutations have also been associated with a more severe phenotype referred to as developmental and epileptic encephalopathy (KCNQ2-DEE) [7,8,9,10,11,12,13]. The clinical phenotype spectrum varies from early neonatal self-limited seizures and normal neurodevelopmental to neonatal onset intractable seizures with severe developmental delays with or, unusually, without convulsions [8,9,11,12,14,15,16]. Early diagnosis and effective treatment are crucial, and, for outcome predictions, an accurate classification of *KCNQ2* missense pathogenic variants is needed [11]. The effective treatment and the genetic counseling involved in each case must start with the correct identification of *KCNQ2* pathogenic variants. To address this, computational tools, particularly machine learning (ML), have been applied to predict the pathogenicity of protein variants [17]. 

Despite relative success in differentiating between severe disease-causing and tolerated variants in some genes, existing tools are inconsistent in their predictions [18,19,20], with the accuracy shown to be gene-dependent [21,22], and do not perform well for the *KCNQ2* gene [13,23]. Existing variant effect predictors (VEPs) present limitations in accurately classifying *KCNQ2* missense variants and do not assess the severity of the associated disease. Notably, these tools tend to mislabel more than 10% of *KCNQ2* pathogenic variants as tolerated and more than 10% of tolerated variants as pathogenic. 

This study presents a significant advancement in the accurate classification of *KCNQ2* missense variants, overcoming the limitations of existing tools. Accessible at https://channels.bcb.eus/ (accessed on 21 June 2023), the developed MLe-*KCNQ2* model, with its high sensitivity and specificity, provides a valuable resource for clinicians and researchers. Notably, this model can assist in anticipating the severity of *KCNQ2* pathogenic variants, a critical aspect for clinical decision making and for early initiation of intervention programs. Our findings highlight the importance of considering gene-specific factors in developing accurate ML models for variant classification. We find that leveraging a novel Variant Frequency Index (VFI) to capture allelic frequency variation and structural information provides a robust framework for enhancing the classification accuracy in the context of *KCNQ2* missense variants. The methodology of this study, rooted in gene-specific considerations and ML techniques, establishes a transferable framework with broad implications for genetic variant classification and clinical prognosis.

## 2. Results 

We collected a well-balanced set comprising 554 *KCNQ2* variants, where 285 were characterized by a high likelihood of being tolerated due to their presence in populations without manifestation of neurological symptoms, or being annotated as “benign” in different databases (Appendix A). Figure 1 illustrates the spatial distribution of these tolerated variants (green) along 269 variants labeled as “pathogenic” (red) within the K_V_7.2 3D structure modeled by AlphaFold2 [24]. Although most databases classify variants as “pathogenic” or “benign”, we refer to these “benign” variants as “tolerated” to emphasize the difference between variants which do not cause pathologies in patients and pathological variants that are related to benign familial neonatal epilepsy or more severe manifestations.

We assessed the predictive capabilities of nearly 80 computational tools using this dataset (Figure 2 and Appendix A) by plotting the percentage of correct assignments for both tolerated and pathogenic variants. Notably, 35 tools annotated more than 90% of the pathogenic variants correctly, whereas only 5 achieved a similar performance for tolerated variants. Remarkably, none of the tools concurrently annotated more than 90% of the tolerated and pathogenic variants correctly. In fact, the most sensitive tools for detecting pathogenic variants (KCNQ2_Index [23], DeMag [26], and MAVERICK [27]) are the least sensitive for detecting tolerated ones, while the most sensitive for detecting tolerated variants are the least sensitive for detecting pathogenic ones (CPT1 [28] and GEMME [29]). Subsequently, we conducted an in-depth investigation into how various ML algorithms and descriptors could enhance the concurrent improvement in accuracy scores for both tolerated and pathogenic variants.

### 2.1. VFI Score Comparison

We begin our analysis by examining the impact that different definitions of the VFI score to capture the effects of allelic variation tolerance on performance. Multiple definitions can be derived by adjusting the standard deviation of the Gaussian kernel, but determining the optimal definition for the given problem is challenging *a priori*. To ascertain the optimal choice of parameters and to understand the sensitivity of the models’ performance to this specific value, we trained several algorithm versions. Each version utilized a VFI score calculated with Gaussian kernels of varying standard deviations. The results suggest that the introduction of additional noise by using wider kernels did not degrade the performance significantly (Table 1). There was a slight trend towards a lower performance as the width of the Gaussian kernels was increased, which means that as the weight assigned to positions further away from the central amino acid increases, the performance tends to degrade slightly. Nevertheless, concentrating most of the weight on only a few amino acids around the central one comes at the expense of a less statistically robust score, potentially hindering generalization to unseen data due to the smaller amount of observations used to derive it. 

Regarding the choice of ML model, most tested algorithms demonstrated very similar performances, with differences in the AUC-ROC scores of approximately 0.01–0.03 between them. The only exception to this pattern was the naive Bayes algorithm, as it consistently exhibited a significantly lower performance across all tests. Despite the similarity in performance, certain algorithms tended to outperform others in most instances. In particular, logistic regression, support vector machine, random forest and gradient boosting algorithms appeared to be the most suitable models for these types of tasks.

An interesting observation is that, in the particular case of *KCNQ2*, the pLDDT score generated by AlphaFold2 is highly correlated with the VFI scores obtained from allele frequency data, and when plotted against sequence positions, the pattern generated is also similar to the VFI score, as can be seen in Figure 3A. However, VFI has a higher resolution compared to pLDDT, meaning that it exhibits a higher position-dependent variance, while the overall pattern is very similar. Thus, we also tested whether replacing the VFI score with pLDDT would have a significant impact on the model performance by training another set of models using only pLDDT instead of VFI.

Models trained using pLDDT slightly underperformed compared to VFI models (Table 1), but still achieved comparable results without utilizing allelic variation data. This aspect may be desirable in cases of significant data scarcity. However, when VFI is employed as a feature, pLDDT provides little additional information and is often disregarded by the feature selection algorithm as a result of the high correlation between both features. Considering these results, we selected the models trained using a VFI score with kernels of a standard deviation of 2. Although the margin is small, these models achieve the highest scores without using narrow kernels that would reduce their statistical robustness. 

### 2.2. Model Analysis

To better understand the inner workings of our models, we examined the features that had the biggest impact on their predictions, as well as the variants that were most likely to be misclassified during cross-validation. For a detailed examination of individual features, our focus was on the top-performing model among those trained—the logistic regression model utilizing a VFI score with a standard deviation of 2. We quantified feature importance through Shapley values, employing the SHAP package [30]. Shapley values offer a reliable measure of feature importance with desirable properties, including a solid theoretical foundation rooted in game theory, and model agnosticism, enabling us to compare importance values across different models.

As depicted in Figure 3C, where the mean Shapley values are plotted, the VFI score is the most relevant feature in determining the prediction of the classifier. Although individual contributions from the remaining features are relatively modest, their collective impact remains substantial in shaping the overall model outcome. This observation can be partially attributed to the categorical nature of many features, which implies that they do not have a significant effect on the majority of examples’ outcomes. Instead, their role can be understood as adjustments to the VFI-based predictions, depending on the specific properties of the mutation.

This effect can be better observed in the beeswarm graph in Figure 3B, where we can observe that, when the variant belongs to this category, many of these features play a significant factor. However, given that the features are zero for the majority of examples and, consequently, the SHAP values are very close to zero in such cases, the overall influence of these features turns out to be relatively low when averaged over all variants. Some examples of this are the p_to_np feature, which encodes whether the mutation transforms a polar amino acid into a non-polar one, or the unknown_function feature, which encodes whether the mutation occurs in the domain of the protein with an unknown function.

Although the examined metrics already give us a good estimation of the performance of our algorithms, it is also of interest to investigate the cases where the algorithm fails. For this purpose, we define an error as a case in which a variant present in the test set during a cross-validation iteration is incorrectly classified. As the cross-validation procedure was repeated 25 times, each variant will appear in the test set exactly 25 times, with different sets of variants in the training set. This gives us a good method to estimate which variants the algorithm is most prone to misclassify by computing the ratio of total misclassifications and the total 25 predictions for every variant. For 16 variants, the error rate is above 50%: 6 pathogenic variants are predicted as tolerated and 10 tolerated variants are predicted as pathogenic (Figure 4). Each of these variants has its own particular factors that make them challenging to predict. For example, G256E is classified as a tolerated variant, yet other variants at the same position, including G256W, G256V and G256R, are classified as pathogenic. As a result, the VFI score is very low in this region and the model makes confident predictions based on this factor. Situations like these, where the pathogenicity of different variants in close regions is determined by intricate chemical differences, may be challenging for models like this, as they rely on small datasets that are not able to capture these relationships.

### 2.3. Ensemble Model

Although the models trained with different VFI scores already obtain a satisfactory performance, we also examined whether ensemble models combining the predictions made by different algorithms could improve the overall performance further (Table 2). We used the algorithms trained with the best-performing version of the VFI score, in this case, those with a standard deviation of two (σ = 2), and tested multiple ensemble strategies to combine these algorithms.

There was a slight improvement in the accuracy for the ensemble models. In addition, we also found that a single soft voting strategy, where the scores obtained by individual models are aggregated through simple averaging, obtained a better performance than more stacked classifiers trained with the scores of the models as features. We also observed that combining more than four models in the ensemble caused the performance to regress, suggesting that the potential benefits of combining multiple models is limited. When evaluating these models on the test set, we also observed a similarly high performance, suggesting that the metrics estimated through cross-validation can be generalized to completely unseen data. Despite the limited size of the test set, these results, along with the cross-validation estimates, validate the robustness of the model performance metrics.

### 2.4. Severity Prediction

Since the severity, future developments and clinical consequences are dramatically different, from a clinical point of view, once a variant is identified as pathogenic, the correct identification of pathogenic variants specifically associated with developmental and epileptic encephalopathy (DEE) is also critical, along with their differentiation from pathogenic variants related to self-limited neonatal epilepsy (SeLNE). We compiled a dataset (Appendix A) comprising a subset of pathogenic variants, which were subsequently classified as either severe or benign based on a bibliographical review (Figure 5). From this dataset, we developed a model specifically tailored for the classification of pathogenic variants into these two subgroups.

The training of this model followed the same steps outlined in the methodology section. The only necessary adjustment for this learning task was the exclusion of certain features, specifically the initial and final amino acids involved in the mutation. This adjustment was made because we observed that models trained with these features had a tendency to overfit to the training data, essentially memorizing specific mutation patterns within the training set but failing to generalize beyond these examples. To address this, we entirely removed these features. We observed that, while the training error increased, the cross-validation error decreased, indicating that the inclusion of these features was hindering generalization.

Upon inspecting the trained models, we discovered that the criteria used for classifying variants as benign or severe differed significantly from the criteria used to determine their pathogenicity. SHAP scores for the features used by the best-performing algorithm revealed that no single feature dominated over all others in this case (Figure 5). Instead, the algorithms relied on contextual information, such as whether the variant interacts with calmodulin or is located in the selectivity filter of the channel. Interestingly, while the VFI score was highly valuable for correctly classifying variants as pathogenic or tolerable, it seemed to provide little information in this case. In contrast, the pLDDT score appeared to remain somewhat informative.

Following the strategy implemented for pathogenicity classification, individual algorithms trained with different features and hyperparameters were aggregated through an ensemble model that averaged the predicted scores of a subset of algorithms, selected to maximize the cross-validation score of the ensemble model. The performance metrics for the ensemble model, assessed through cross-validation on the independent test set following the same evaluation strategy as the pathogenicity prediction model, are presented in Table 3.

When evaluating the model through cross-validation, it becomes apparent that the model exhibits a discernible bias toward the severe class. This bias is a consequence of the dataset’s imbalance, which, in this instance, could not be rectified by incorporating supplementary datasets. To correct this bias, the threshold used to predict a label was modified to optimize the balanced accuracy of the model on the training set, and this optimized threshold was then used to assign labels to variants in the test set. As a result, we can see that predictions for the test set do not show a significant bias towards either class.

However, it is noteworthy that the AUC-ROC for the test set falls slightly below the expected value derived from cross-validation. Unlike the previous model, where significant deviation between both performance estimations was not observed, potential data leakage, stemming from the fact that the training data used for cross-validation were also employed for feature and hyperparameter selection, might be contributing to a slight overestimation of the model’s performance.

Overall, the model achieves a reasonable performance on a very challenging prediction task, but the limitations arising from data scarcity, class imbalances and a lack of highly informative features are clear. Despite these limitations, our model demonstrates the potential of ML models to provide valuable Appendix A for the prognosis, complementing traditional medical knowledge.

## 3. Discussion

The *KCNQ2* gene encodes for the K_V_7.2 K^+^ channel, which modulates neuronal excitability. Mutations in the *KCNQ2* gene cause highly diverse phenotypes ranging from normal, familial neonatal seizures to early onset developmental and epileptic encephalopathy. SeLNE is characterized by seizures in newborns that begin between 2 and 8 days of life and usually spontaneously vanish within 1 to 6–12 months. Other mutations in the same gene can induce DEE, characterized by more severe seizures and moderate to profound motor impairment and intellectual disability [1]. Since individualized therapies and genetic counseling are needed in these patients and families, a rapid and correct identification of the pathogenicity of the variant is crucial.

The aim of this work was to develop a tool for the prognosis of *KCNQ2* gene missense variants with practical applicability in a clinical setting. In the initial phase, we constructed an ensemble model capable of distinguishing between tolerated and deleterious *KCNQ2* variants, accurately predicting over 95% of both tolerated and pathogenic variants concurrently. We established a user-friendly web page interface (https://channels.bcb.eus/ accessed on 21 June 2023) for accessing predictions on any amino acid substitution in the K_V_7.2 protein or any single nucleotide variant in the *KCNQ2* gene. Additionally, predictions for all substitutions on the K_V_7.2 protein can be downloaded. Our model is not intended to replace clinical judgment but rather to inform and complement clinical decision making through objective and quantifiable data.

### 3.1. Comparison with Other Tools

MLe-KCNQ2 exhibits superior accuracy in classifying *KCNQ2* missense variants compared to over 80 tested tools. Among the unsupervised methods not trained on disease-causing variants [31,32], a noteworthy performance was observed in models relying on multiple sequence alignment (MSA) of homologous and orthologous K_V_7 proteins, such as GEME [29] and EVE [33]. This underscores the significance of sequence conservation as a crucial feature for pathogenicity predictions [34,35,36]. An alternative strategy is to train models using deep mutational scanning data from a few proteins to generate cross-protein transfer models for variant interpretation of unseen proteins, as is achieved in CPT-1 [28]. More complex artificial intelligence methods could also be employed, for example, deep learning techniques. Although there are already a number of studies that have worked with neural networks, one major drawback of deep learning is that it requires an immense amount of data. AlphaMissense [37] uses a model similar to AlphaFold2 [24], which is initially pre-trained for single-chain protein structure predictions and subsequently fine-tuned for variant effect predictions. Another deep learning approach implemented in ESB1b uses a protein language model trained in millions of protein sequences with millions of parameters [38]. Transfer learning offers an opportunity to leverage the power of deep learning in situations where data are limited [4]. This emerging approach may be implemented in the context of variant effect prediction by training a model using data from a well-studied gene (X) and then refining the model with data from a less-studied gene (Y). The resulting model may perform very well on Y because the “lessons” learned in modeling X transfer well to Y [39]. However, these and other computational tools inaccurately predict a significant proportion of tolerated variants at highly conserved positions as pathogenic, often failing to identify truly pathogenic variants at less conserved positions [36].

Certain supervised methods (i.e., trained on clinical data), specifically KCNQ_Index [23] and DeMag [26], which heavily rely on allelic frequency and structural descriptors, respectively, along with VEST4 [40], VARITY [41] and the more recent MAVERICK tool [27], demonstrate a superior performance compared to unsupervised methods in classifying *KCNQ2* missense variants. However, these tools mislabel more than 10% of *KCNQ2* pathogenic variants as tolerated and vice versa, limiting their clinical utility [42,43].

For model training, we tested multiple algorithms proven effective in diverse ML applications, including logistic regression, support vector machine, and tree ensemble models like random forest and gradient boosting methods. No significant performance differences were observed between these algorithms, with simple linear models such as logistic regression slightly outperforming more complex models like gradient boosting algorithms [44]. Crucially, the effectiveness of ML models for pathogenicity prediction hinges on the quality of the features characterizing each variant and the data used for model training.

The availability of high-quality datasets of classified variants [41] is essential for evaluating the performance of any ML method [31]. While most tools rely on datasets from ClinVar [45] or HGMD [46], these datasets exhibit a notable imbalance for *KCNQ2* variants, predominantly comprising pathogenic labels [47]. To mitigate this bias, we meticulously curated a compendium of 554 missense variants with a balance of tolerated (285 variants) and pathogenic (269 variants) variants. Given the incomplete penetrance of *KCNQ2*-related disorders [1], some variants labeled as “tolerated” in our dataset may warrant reevaluation as “pathogenic”.

For *KCNQ2*, we identified the Variant Frequency Index (VFI) score as the most informative feature. This score, designed to indicate tolerance to variation at each position within the amino acid sequence, is derived from variant population data accessible in gnomAD. Additionally, conventional features such as the evolutionary conservation of residues and the functional domains where mutations are located provided valuable information, enhancing the model’s accuracy beyond the primary descriptor. Despite the pivotal role of the VFI score in our primary models, its inclusion was found not to be essential for developing well-performing models, as similar information appeared embedded in the pLDDT score provided by AlphaFold2. In fact, both VFI- and pLDDT-trained models consistently demonstrated sensitivity and specificity scores above 90% throughout cross-validation.

Our final model, designed as an ensemble of different algorithms and trained with varying features and hyperparameters, achieved specificity and sensitivity scores exceeding 95% in both cross-validation and independent test set estimations. This high accuracy indicates the model’s potential utility in clinical decision making.

### 3.2. Clinical Implications

Although MLe-KCNQ2’s predictions provide a robust foundation for discriminating between tolerated and pathogenic variants, clinicians are advised to interpret the output cautiously. As highlighted by Stead [48], predictions, including those from our models and other top-performing tools such as KCNQ2_Index, DeMag, MAVERICK, VARITY, GEMME and CPT1, may not capture every scenario. For instance, none of the models, including the best-performing ones, accurately predicted the D488E mutation as pathogenic. A patient carrying this mutation experiencing seizures two months after birth exhibited delayed development and remained seizure-free for more than four years at the time of the report [49]. This conservative substitution, involving a negatively charged residue replaced by another negatively charged residue, is conserved across several fish species, and removing a large stretch of amino acids containing this residue results in protein variants with similar electrophysiological properties as the wild type [50]. This residue is located in a region predicted by AlphaFold2 to adopt an alpha helical disposition, and is flanked by A501, where the hard-to-predict pathogenic variant A501P maps. Interestingly, this mutation linked with severe encephalopathy has been found to be inherited from healthy parents [51]. The impact of these variants in health suggests that this putative helix performs an essential unknown role. Thus, some variants disrupt functions that are not yet fully understood.

No genome-wide or *KCNQ2*-specific tool has been designed to assist in assessing the severity of a *KCNQ2* variant. In addressing this gap, we investigated the ability of ML to discriminate pathogenic variants according to disease severity [13]. We curated a high-quality dataset with variants confidently labeled as pathogenic_benign or pathogenic_severe by reviewing the literature. Consequently, we developed MLe-KCNQ2, the first supervised model classifying *KCNQ2* pathogenic variants based on severity. 

### 3.3. Phenotypic Discrimination

While the model excels in predicting pathogenicity, challenges arise in discriminating between different phenotypes. Except for pLDDT, which yielded informative results, models trained for this task did not utilize the most effective features for pathogenicity prediction. Instead, they relied on contextual features related to the mutation’s location in the amino acid sequence. The limited data availability for this task, combined with a lack of features to accurately discriminate between severe and benign cases, presents challenges in developing models with an accuracy comparable to pathogenicity prediction models.

Despite these limitations, we achieved a balanced accuracy of approximately 67% on the test set, a comparable performance to that obtained to predict gain or loss of function for potassium [52], sodium or calcium channel variants [53]. While it may not entirely meet the criteria for clinically reliable predictions, it suggests that our model could offer valuable information when combined with additional clinical data. For pathogenic variants in the neuronal sodium channel α1 subunit gene (*SCN1A*), the time of onset has been found to contribute to discriminating between Dravet syndrome and GEFS++ [54]. Another clinical feature likely to contribute to increasing predictive power is the mode of inheritance (*de novo* vs. familial), as inherited cases are often associated with milder phenotypes.

### 3.4. Limitations

Among other limitations, in this work, only *KCNQ2* missense variants are considered, excluding frameshift, premature stops and nonsense variants. The model cannot inform whether a mutation predicted as pathogenic involves gain- or loss-of-function. Another difficulty arises because some pathogenic variants may have a range of penetrance and magnitude of effects [55]; the same mutation can cause no effect in one carrier and SeLNE in another, or can cause different phenotypes even in two carrying brothers. These difficulties make it advisable to consider complementary information. Another limitation derives from variant reclassifications that regularly occur in clinical databases such as ClinVar [4] (see Appendix A). 

Further improvements are likely to arise by updating the VFI metric with additional genomic data from large-scale sequencing efforts [2,56]. The gnomAD database used to compute the VFI index incorporated about 200,000 samples at the time of this analysis and included 412 out of the 5972 theoretically possible missense variants, which represents about 7% coverage. Due to this limited coverage, the chosen resolution for VFI analysis was 31 codons, i.e., a running average with the preceding 15 residues and the following 15 residues. Segments that exhibit fewer missense variants than expected are flagged as genetically intolerant, indicating that mutations within that gene segment tend to be evolutionary excluded from the gene pool and are significantly enriched for pathogenic missense variants [2]. The size of the running average window should decrease with larger datasets, improving the resolution and its usefulness for predicting potential hotspots within a protein.

AlphaFold2’s output offers the most likely structure that will appear in the PDB database. The confidence metrics for the 3D AlphaFold2 proposal for the K_V_7.2 channel structure (pLDDT) turned out to be a very useful feature for pathogenic predictions of missense variants. This parameter evaluates the local distances on all atoms in a model, including validation of stereochemical plausibility [57]. Thus, mobile regions, underrepresented in the PDB database, are expected to score low. Lower values may correspond to disordered or mobile regions, whereas larger values correspond to residues modeled with a high accuracy. Nevertheless, the relationship between this parameter and flexibility should be treated with caution [24]. Our data suggest that this parameter is useful for the identification of sites with different tolerances for missense mutations [37]. 

With the available data becoming more abundant, ML algorithms will systematically generate improved outputs, and new interesting applications are expected to follow. We suggest further work on the dataset for missense variants of *KCNQ2* by incorporating unseen variants from the gnomAD database or recently reported studies. Designing new features for variant characterization, such as the change in the number of hydrogen donor or acceptor sites, would improve classification metrics as proposed in [58]. Advances in protein structure prediction (e.g., AlphaFold2) as well as cryo-EM technologies could lead to the design of more complex 3D features that could lead to a breakthrough in the prediction of variant pathogenicity. 

### 3.5. Conclusions

MLe-KCNQ2 stands as a significant advancement in the clinical diagnosis of *KCNQ2*-related pathologies. Its high specificity and sensitivity, coupled with its potential extension to other genes, position it as a valuable tool for genetic diagnoses and hold promise for unveiling new aspects of pathogenic landscapes within the broader context of genetic research. In addition, an analysis of the pathogenic landscape could help to identify previously unknown domains within the protein sequence and to discover new functions and possible new pathways for treatments. Our model could thus aid clinicians and researchers in interpreting missense mutations in the *KCNQ2* gene, facilitating clinical diagnosis and enabling the early selection of personalized therapies for developmental *KCNQ2*-related disorders.

## 4. Materials and Methods

### 4.1. DATASET

Isoform 1 of the *KCNQ2* gene encodes a transcript with 872 codons, which could result in 7582 single base substitutions, leading to 5972 single amino acid missense variants (excluding 266 substitutions that introduce a stop codon). To date, more than 1400 *KCNQ2* variants have been annotated in the ClinVar database (https://www.ncbi.nlm.nih.gov/clinvar/) (accessed on 21 June 2023), making it an invaluable source of data for the development of algorithms capable of distinguishing neutral and pathogenic variants. Recent work [33,59] has also demonstrated that models trained through unsupervised and self-supervised methods can achieve a remarkable performance, even in the absence of pathogenicity data, where the complete set can then be used for evaluation without circularity issues. However, biases still exist due to the uneven representation of certain families of proteins in the datasets they are trained on, as well as heuristic criteria used to generate labels for self-supervised training. As a result, the accuracy may not be adequate for clinical effectiveness. Therefore, specialized models that account for the unique features of individual genes are necessary [60].

As a first step to produce a high-quality dataset, we investigated RIKEE (https://www.rikee.org) (accessed on 21 June 2023), a high-quality database curated by a panel of highly recognized active experts in the *KCNQ2* pathology field, updated up to 2015. More than 700 *KCNQ2* missense variants are already annotated in ClinVar [45] (i.e., ~10% of potential missense variants), which are distributed according to their pathogenicity, thus serving as an excellent source of information for the development of predictive models. Additional data were gathered from HGMD [61] and the Global Variome Shared (LOVD) databases [62], supplemented by an exhaustive bibliographic search. In cases where variants were found in multiple databases, an additional filter was imposed to ensure that variants with conflicting interpretations or of uncertain significance were not included. The Final_Dataset contained 269 variants labeled as “pathogenic”. A total of 92 were found in RIKEE, 37 in LOVD3, 81 in ClinVar and 59 in the HGMD database. A total of 285 variants were labeled as “tolerated”, and 1 was found in RIKEE, 6 in LOVD3 and 9 in ClinVar labeled as “Benign” or “Benign/Likely benign”. The remaining variants were found in the non-neuronal dataset of GnomAD [56] with an extrapolated prevalence of more than 3 per million. In sum, the Final_Dataset consisted of 285 tolerated and 269 pathogenic variants distributed along all K_V_7.2 functional domains (Figure 1). After that, clinical labels were binarized, with tolerated and pathogenic variants equal to 0 and 1, respectively.

The “training dataset” included 206 variants labeled as “pathogenic” from the “Final_Dataset”. Due to the close evolutionary proximity between humans and primates, protein variants that are tolerated in these species are likely to also be tolerated in humans [25,63]. Following this criterion, we categorized *KCNQ2* variants commonly found in primates as tolerated in humans and incorporated them into the training of our predictive models. 

Because the examples in the primate dataset were labeled based on heuristics and thus were more prone to labeling errors, we constructed the test set for the final performance estimation purely from the data obtained through ClinVar and other databases in order to generate an unbiased test set for a reliable estimation of the true performance of the model.

We reviewed the literature for the description of 274 pathogenic variants. Those that were described as “self-limiting” were labeled as “pathogenic_benign”, whereas those characterized by developmental and/or encephalopathy were labeled as “pathogenic_severe”. The final_severity_set included 62 variants labeled as “benign”, 32 labeled as “benign/severe” when there were reports for both benign and severe phenotypes, and 180 labeled as “severe”. For model building, “benign/severe” and “severe” classes were combined into the “severe” label.

### 4.2. Variant Characterization

Variants were characterized using a set of features related to physico-chemical changes and biological-evolutive and structural features. The features related to physico-chemical changes were assigned based on the initial and final amino acids involved in the mutation. For each variant, the charge, hydrophobicity, molecular weight, polarity, aromaticity and mean solvent accessibility of the initial and final amino acids were computed, and the differences between the initial and final values were used as features. Given that some pathogenic mutations in the dataset are located at the start codon, we also included a feature indicating whether the mutation occurred in the first position, thus allowing the model to easily classify these cases.

Biological-evolutive features included the topological and functional domains affected by the mutation, as well as the evolutionary conservation value across species of the substituted residue, which was computed as the Shannon entropy for every position in the multiple sequence alignment [64]. In addition, information about the empirically observed human variation in the *KCNQ2* gene was incorporated through a descriptor inspired by the missense tolerance ratio score (MTR) [2], which is a measure of the tolerance to missense variation within a region of a gene, computed based on the number of variants observed in aggregated datasets such as ExAC and GnomAD [65]. Rather than only using data available for every amino acid in the sequence, MTR performs these variant counts in windows of 31 amino acids to prevent excessive variability in regions where the number of observed variants is low.

Despite its effectiveness, we observed a few limitations in the original MTR score definition. A notable drawback arises in cases of methionine and tryptophan, which lack synonymous mutations. In such instances, the MTR score is automatically reduced to 1, resulting in the loss of information regarding positions associated with these amino acids. We have devised an alternative metric, which we call the Variant Frequency Index (VFI). Initially, we calculate the position-wise tolerance of an amino acid sequence using the formula:$$f=\frac{\left[missenseobserved\right]}{N\cdot \left[missensepossible\right]}$$
where *N* is the total number of sequences in the dataset and [*missenseobserved*] and [*missensepossible*] denote the number of observed and possible missense variants at that position, respectively. Subsequently, the position-wise missense variant frequency is constrained between 0 and 1 through a rational function:$$F=\frac{f}{f+\alpha }$$
where α serves as a coefficient that allows modulation of the curvature of the score by establishing the threshold at which *F* = 0.5. The data for this score were calculated based on data from versions 2 and 3 of GnomAD, merged in such a way that redundant data were discarded.

Similar to the approach taken by MTR, instead of directly using the tolerance as the score, we opted to aggregate information from neighboring positions in the amino acid sequence. However, we aimed to enhance the resolution of the score by assigning greater weights to positions closer to the central position during aggregation. In this context, we interpreted the sequence-position-dependent tolerance score (*F*) as a noisy signal of the true tolerance. Our goal was to denoise this signal by aggregating local information. This is commonly accomplished by applying a discrete convolution over the sequence, where the form of the kernel determines the way in which the convolution aggregates information, resulting in a score of the form:$$VFI\left(n\right)=\sum_{j=1}^{m}K\left(j\right)\cdot F\left(n-j+m/2\right)$$

The sliding window may be interpreted as a convolution with a rectangular kernel, but other functional forms can be used. To compute the *VFI* score, we made use of the Gaussian kernel, which is one of the most commonly used kernels and also satisfies the desired property of assigning more weight to positions closer in the sequence. The specific form of the kernel is thus given by:$$K\left(n\right)=\frac{1}{\sqrt{2\pi \sigma^{2}}}e^{\frac{-n^{2}}{2\sigma^{2}}}$$

Finally, we also made use of structural features, including the location within the structural landscape of *KCNQ2* and the secondary structure of the mutated amino acid, which was predicted through the PROTEUS2 metaserver [66]. We also made use of the pLDDT score provided by the AlphaFold2 [24] prediction, which is a measure of the confidence or reliability of the predicted protein structure. Motivated by the similarity between the profiles of the pLDDT and VFI scores for the *KCNQ2* gene, we explored whether this score could serve as a valid alternative to VFI-like measures, whose accuracy is limited by data availability.

### 4.3. Model Definition, Training and Optimization

To find the best possible model for variant effect prediction, multiple ML algorithms were tested. In previous studies, logistic regression, support vector machine and random forest have proven to be useful in variant pathogenicity predictions, as well as in unbalanced class scenarios [44]. In addition, other powerful algorithms such as K-nearest neighbors, linear discriminant analysis, gradient boosting and Gaussian processes were also considered as potential candidates. For gradient boosting in particular, multiple implementations with slight differences among them exist. In this work, three different implementations of this algorithm were tested, namely LightGBM, CatBoost and XGBoost.

For each of these algorithms, a pipeline for optimal feature and hyperparameter selection was implemented. First, optimal features were selected based on a forward greedy search approach, where features were iteratively added one at a time, selecting the feature that optimized the performance of the model at each step. The search was stopped when the performance of the algorithm did not improve for over 10 steps, and the subset of features that achieved the maximum performance throughout the search process was selected for further analysis. Second, optimal hyperparameters were selected based on a random grid search over a predefined search space, where a random set of hyperparameter combinations was tested, and the subset with the highest performance was selected. The search space for each hyperparameter was selected based on a visual inspection of validation curves for individual hyperparameters. For both the feature and hyperparameter selection steps, the performance of the models was evaluated by performing a stratified 5-fold cross-validation and measuring the AUC-ROC averaged over each split.

After selecting the optimal features and hyperparameters for each algorithm, their performance was estimated by repeated 5-fold cross-validation, where the cross-validation procedure was repeated 25 times by shuffling the dataset, thus generating different splits at each iteration. Performance metrics were measured for each split, and the final performance was evaluated by considering the average and the standard deviation of each of these metrics over all splits. This allowed us to obtain more robust estimates of the performance of our models, as they were evaluated under many possible data splits.

It is important to note that the performance estimates obtained through repeated cross-validation are biased by the fact that the evaluation was performed on the same data that were used for feature and hyperparameter selection, which may lead to overestimation of the model’s capability to generalize to unseen data. To validate that our estimates are reliable, we evaluated our models on the unseen test set we previously set aside (89 variants) after re-fitting each model to all training data. In every case, the performance metrics on the test set fit within the ranges estimated by repeated cross-validation, suggesting that despite the bias, this procedure provides reliable estimates of the true performance of the models for this problem.

In order to enhance the overall efficiency and mitigate bias in individual models, we implemented ensemble models that aggregate the predictions from these models into a unified prediction. Various strategies for model ensembling exist, such as hard voting, where the most commonly predicted label is selected, and soft voting, which involves averaging the predicted probabilities from the individual models and selecting the class with the highest average probability as the final prediction. An alternative approach is to train an additional algorithm that takes the predicted probabilities from multiple models for each variant as inputs and learns to classify variants based on these probabilities. 

To determine the most effective ensembling approach for this problem, we adopted a similar philosophy to that used to find optimal individual models. Employing a greedy approach, we iteratively added models to the ensembles, and then selected the subset of models that optimized the ensemble’s performance, measured through 5-fold cross-validation. Ultimately, the ensembling strategy that exhibited the best performance was chosen as the final model.

## Figures and Tables

**Figure 1 ijms-25-02910-f001:**
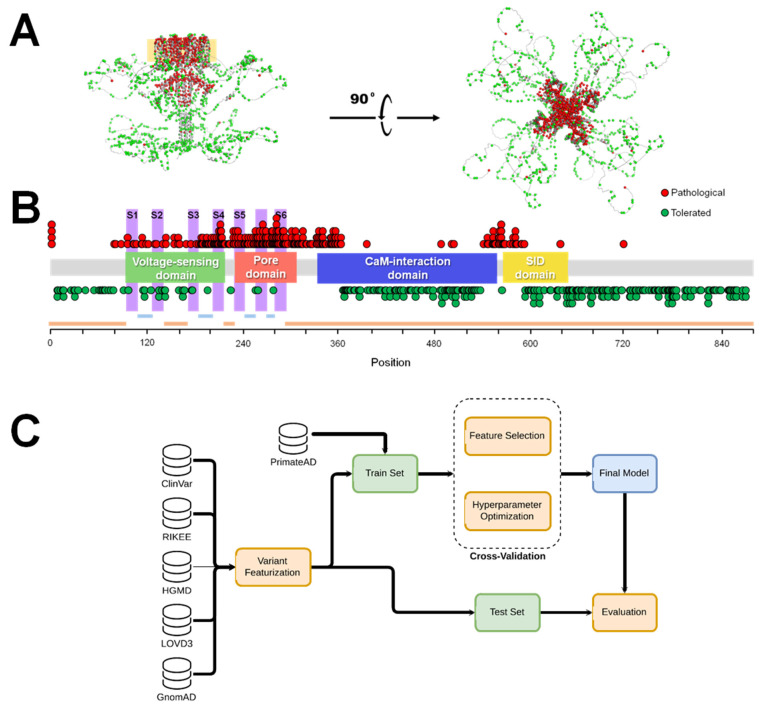
Graphical distribution of the collected variants. (**A**) Distribution on the AlphaFold2 predicted structure of a K_V_7.2 channel tetramer, lateral (left) and upper view (right). The membrane is indicated with an orange square in the lateral view. Pathogenic and tolerated variants are colored in red and green, respectively. (**B**) Distribution along the main functional domains of the channel. Pathogenic variants are found most abundantly in the voltage-sensing domain (VSD), the pore, the first residues of the calmodulin (CaM) interaction domain and the transition between helices B of the CaM interaction domain and helix C of the subunit interaction domain (SID). Information about extracellular (light blue), transmembrane (pink) and cytoplasmic (orange) segments is shown at the bottom. (**C**) Overview of the study workflow. For phenotypic discrimination, a “Final_Dataset” was assembled from RIKEE (92 P, 1 T), LOVD3 (37 P, 6 T), ClinVar (81 P, 9 T), HGMD (59 P), and GenomAD (269 T), where P stands for “pathogenic” and T for “tolerated”. The training set contains 206 P variants from the “Final_Dataset”, and 10 T variants from ClinVar and RIKEE. The test set consisted of 27 T and 62 P variants from the “Final_Dataset”, distinct from the training set. To balance the representation of T and P variants during training, 208 variants from PrimateAD [25] (152 also present in GenomAD), labeled as T, were incorporated into the training set.

**Figure 2 ijms-25-02910-f002:**
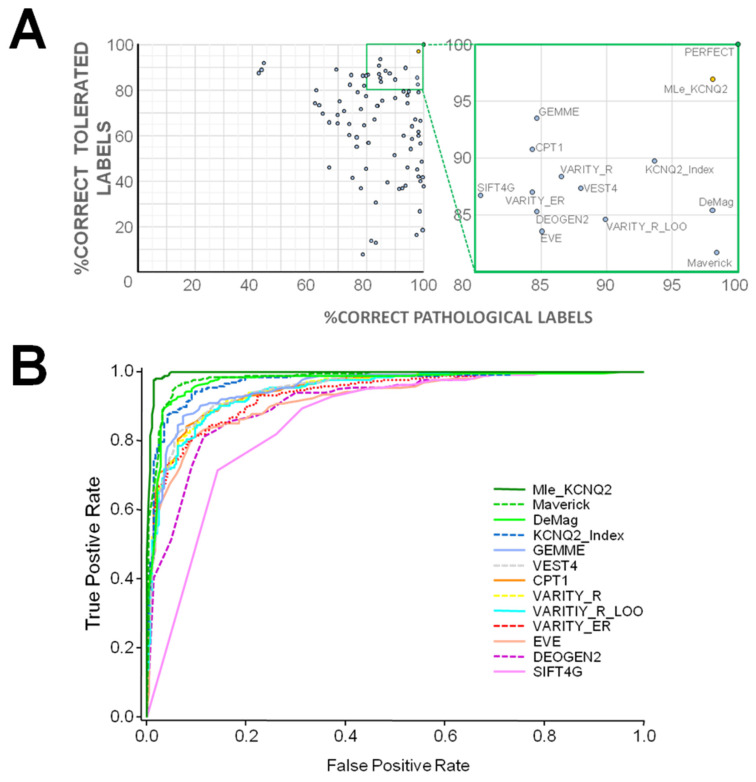
(**A**) Representation of the percentage of correctly assigned labels for tolerated and pathogenic variants by different tools. A perfect score will be placed on the top right corner (green filled circle). The yellow filled circle corresponds to the assessment of the MLe-KCNQ2 tool described in this paper. (Right) Zoom in of the area with correct assignments above 80%, labeling each corresponding tool. (**B**) Receiver operating characteristic (ROC) curves showing the relationship between the observed true and false positive rate for the best performing models highlighted in panel (**A**). Tables with the values can be found in the Appendix A.

**Figure 3 ijms-25-02910-f003:**
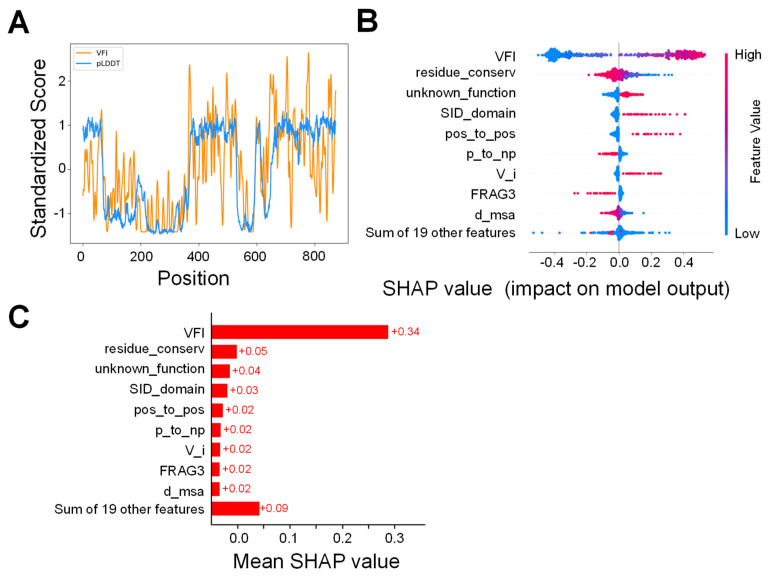
(**A**) Comparison of VFI scores, which encode the tolerance to variants in each position in the protein sequence, and the pLDDT score generated by AlphaFold2. (**B**) Beeswarm plot of the SHAP scores for each feature used by the best performing algorithm. Each point represents the value of the feature for a single variant, and the value of the feature is represented in a color scale. The SHAP values indicate the extent to which they influence the prediction of the model. Positive SHAP values indicate that this value of the feature influences the algorithm to make a positive prediction, whereas negative values have the opposite effect. (**C**) Absolute values of the SHAP scores for each feature, averaged over all samples in the training set. Higher values indicate a larger average influence on the prediction of the model.

**Figure 4 ijms-25-02910-f004:**
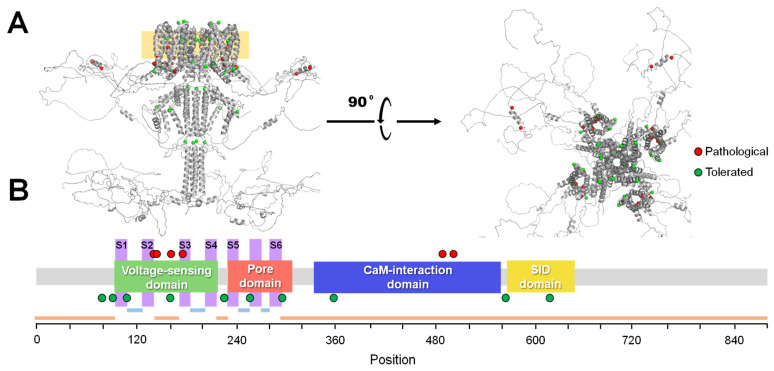
Map of variants with the highest ratio of errors during cross-validation averaged over all trained algorithms. From highest to lower error, the variants were D488E (P), G256E (T), A501P (P), W91G (T), V225I (T), H357R (T), V108M (T), E140Q (P), A295S (T), V175L (P), V564L (T), R160Q (T), Q78K (T), L161P (P), P617S (T), R144Q (P). P stands for pathogenic, erroneously labeled as tolerated, and T stands for tolerated, erroneously labeled as pathogenic. (**A**) Distribution of these variants on the AlphaFold2-predicted structure of a K_V_7.2 channel tetramer, lateral (left) and upper view (right). The membrane is indicated with an orange square in the lateral view. Mislabeled pathogenic and tolerated variants are colored in red and green, respectively. (**B**) Distribution of the mislabeled variants along the main functional domains of the channel. Information about extracellular (light blue), transmembrane (pink) and cytoplasmic (orange) segments is also shown.

**Figure 5 ijms-25-02910-f005:**
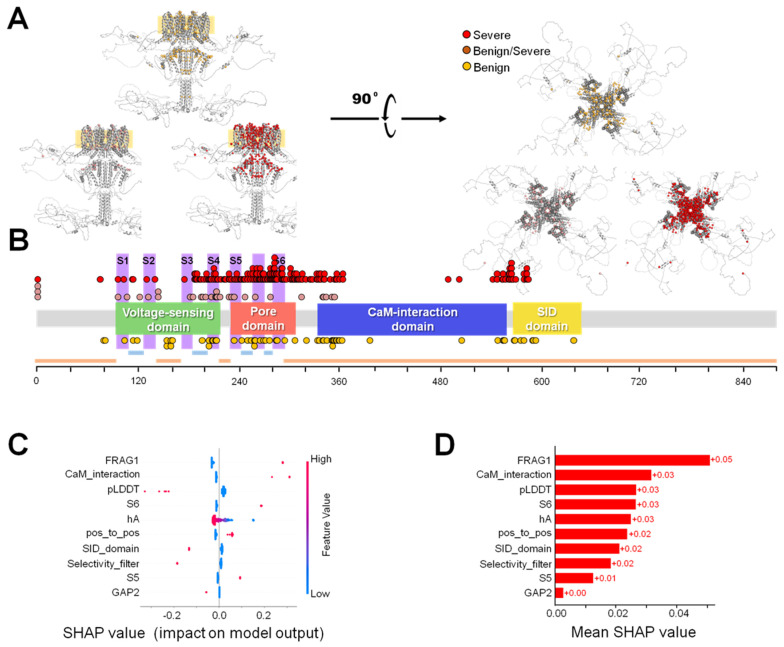
Map of pathogenic variants according to severity. (**A**) Distribution of the collected variants on the AlphaFold2-predicted structure of a K_V_7.2 channel tetramer, lateral (left) and upper view (right). The membrane is indicated with an orange square in the lateral view. Severe, benign/severe and benign variants are colored in red, dark purple and orange, respectively. (**B**) Distribution of these variants along the main functional domains of the channel. Information about extracellular (light blue), transmembrane (pink) and cytoplasmic (orange) segments is also shown. (**C**) Beeswarm plot of the SHAP scores for each feature used by the best performing algorithm for severity discrimination. Each point represents the value of the feature for a single variant, and the value of the feature is represented in a color scale. The SHAP values indicate the extent of influence in the prediction of the model. Positive SHAP values indicate a trend to make a positive prediction, whereas negative values have the opposite effect. (**D**) Absolute values of the SHAP scores for each feature, averaged over all samples in the training set. Higher values indicate a larger influence on the prediction of the model.

**Table 1 ijms-25-02910-t001:** Performance of trained models for discrimination between tolerated and pathogenic variants. Performance scores for the best algorithm trained with each type of VFI score and pLDDT. LR stands for logistic regression, SVM for support vector machine and RF for random forest.

Algorithm Used		AUC-ROC		Balanced Accuracy		Sensitivity		Specificity	
		Mean	Sd	Mean	Sd	Mean	Sd	Mean	Sd
VFI (σ = 2)	RF	0.987	0.009	0.946	0.022	0.943	0.034	0.951	0.036
VFI (σ = 3)	LR	0.985	0.008	0.939	0.021	0.956	0.033	0.924	0.037
VFI (σ = 4)	RF	0.982	0.011	0.930	0.026	0.937	0.040	0.923	0.040
pLDDT	SVM	0.973	0.015	0.920	0.026	0.908	0.039	0.931	0.041

**Table 2 ijms-25-02910-t002:** Ensemble models for tolerated vs. pathogenic discrimination. Cross-validation and test set performance scores for ensemble models, trained by aggregating the best subset of models according to the findings in Table 1. The single soft voting strategy obtained better performance. Combining more than four models in the ensemble caused the performance to regress. A similar performance was observed when using the test set.

	AUC-ROC		Balanced Accuracy		Sensitivity		Specificity	
	Mean	Sd	Mean	Sd	Mean	Sd	Mean	Sd
Cross-validation	0.993	0.005	0.961	0.021	0.966	0.031	0.956	0.030
Test set	0.995	-	0.991	-	0.983	-	1.000	-

**Table 3 ijms-25-02910-t003:** Severity model. Cross-validation and test set performance scores for the ensemble model trained for the prediction of severity. Cross-validation scores are reported with predictions made based on a prediction threshold of 0.5. Test set scores are reported after the threshold was modified to optimize the balanced accuracy of the model on the training set, which balances the sensitivity and specificity of the model.

	AUC-ROC		Balanced Accuracy		Sensitivity		Specificity	
	Mean	Sd	Mean	Sd	Mean	Sd	Mean	Sd
Cross-validation	0.732	0.071	0.603	0.057	0.879	0.068	0.327	0.135
Test set	0.675	-	0.667	-	0.683	-	0.652	-

## Data Availability

The data and code generated during this study will be made available on GitHub at https://github.com/BilbaoComputationalBiophysics/MLe_KCNQ2 (accessed on 21 June 2023) at the time of publication.

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
