# Peer review of "MLe-KCNQ2: An Artificial Intelligence Model for the Prognosis of Missense KCNQ2 Gene Variants"

_ijms, 2024, doi:10.3390/ijms25052910_

Round 1

Reviewer 1 Report

Comments and Suggestions for Authors

This is a great manuscript and describes the tool that is very much needed. I also like to focus on epilepsy-related genes and I feel the potential that this tool can be extended to other genetic conditions.

I have a few comments to improve the paper:

1. I missed a workflow description. It is there, but I encourage the authors to make a flowchart showing the points for data entries, the different tools, the order of tools, the process and the outcome. That would massively improve the understanding of the manuscript.

2. The same goes for comparing their tool to other existing tools. The authors have made comparisons and described them in the text, but better graphs and figures with normal statistical comparison would help the readers see the value of their tool. This is the presentation of the results and maybe a bit deeper statistical comparison.

3. I also missed a proposal on how this tool can be implemented in clinical settings and what is needed to get it to clinical trials or real-life applications. Sure, the data underlying the work are real-life data, and this is a really strong input. But maybe the proposal for independent clinical study would be in place with the design (double-blinded etc.) requirements.

4. What are the other potential genetic conditions where this tool can be used?

5. I this tool more for diagnostics or for risk prediction?

Author Response

Referee 1

  1. I missed a workflow description. It is there, but I encourage the authors to make a flowchart showing the points for data entries, the different tools, the order of tools, the process and the outcome. That would massively improve the understanding of the manuscript.

This is a great suggestion, and we agree that it will make the paper more readable. We have therefore produced a figure with the workflow employed included as a panel in Figure 1.

  1. The same goes for comparing their tool to other existing tools. The authors have made comparisons and described them in the text, but better graphs and figures with normal statistical comparison would help the readers see the value of their tool. This is the presentation of the results and maybe a bit deeper statistical comparison.

We thank the referee for this suggestion. We have included an additional panel in figure 2 plotting the receiver operating characteristic (ROC) curves showing the relationship between false and true positive rate for the best performing tools analyzed. In addition, we have added a supplementary file with the AU-ROC computed for the tools analyzed in Figure 2.

  1. I also missed a proposal on how this tool can be implemented in clinical settings and what is needed to get it to clinical trials or real-life applications. Sure, the data underlying the work are real-life data, and this is a really strong input. But maybe the proposal for independent clinical study would be in place with the design (double-blinded etc.) requirements.

We agree with the referee that addressing this issue is important. We have included a section 3.5, which addressed this topic:

MLe-KCNQ2  is.. “a valuable tool for genetic diagnoses and hold promise for unveiling new aspects of pathogenic landscapes within the broader context of genetic research. In addition, the analysis of the pathogenic landscape could help identifying previously unknown domains within the protein sequence, and to discover new functions and possible new pathways for treatments. Our model could thus aid clinicians and researchers in interpreting missense mutations in the KCNQ2 gene, facilitating clinical diagnosis, and enabling the early selection of personalized therapies for developmental KCNQ2-related disorders.”

  1. What are the other potential genetic conditions where this tool can be used? 5. Is this tool more for diagnostics or for risk prediction?

The model we have developed is tailored specifically for missense variants of the KCNQ2 gene. As such, its primary purpose is to predict variants within this gene associated with developmental delay and various epilepsies. While the methodology used to develop this gene-specific model may be applicable to other genes, it's important to emphasize that the model's effectiveness is optimized within this niche.

Therefore, in our opinion, we recommend limiting its use to variants within the KCNQ2 gene to fully leverage its capabilities. This is precisely why our model performs better in KCNQ2 than other "general" tools.

Reviewer 2 Report

Comments and Suggestions for Authors

Dear Editor,

This article discusses the development of an artificial intelligence model called MLe-KCNQ2 to predict the prognosis of KCNQ2 gene variants. The KCNQ2 gene encodes for the KV7.2 K+ channel, which modulates neuronal excitability. KCNQ2 missense variants are linked to developmental delay and various epilepsies, including self-limited benign familial neonatal epilepsy and epileptic encephalopathy.

The authors describe that current tools exhibit a tendency to overestimate deleterious mutations, overlooking tolerated variants (these tools tend to mislabel more than 10% of KCNQ2 pathogenic variants as "tolerated", or more than 10% of tolerated variants as "pathogenic").

For this reason, they propose using machine learning (ML) to predict the pathogenicity of protein variants. Thus, they propose the ensemble model called MLe-KCNQ2. Their proposal still has a complementary tool with a user-friendly web interface available at https://channels.bcb.eus.

To evaluate the method, they collected 554 KCNQ2 variants, where 285 were “benign” and 269 variants were “pathogenic”, and modeled the 3D structure using AlphaFold2.

Here comes my first question to the authors, which confused me when reading the article.

In lines 85-88: you define "benign vs pathogenic" as labels, but in lines 213-216 (and 230): you start using "pathogenic vs tolerated". In line 252, severe or benign become subclasses of pathogenic.

Would it be possible to standardize the labels used? The current version is causing some confusion.

Table 2 indicates that they obtained 100% specificity in the test. However, in Table 3, you obtained high sensitivity and low specificity in the training stage using cross-validation, but the values were close in the test. I didn't understand the modification made to make this happen. Could you describe it better?

Line 143: you affirm that the "pLDDT score generated by AlphaFold2 is highly correlated with the VFI scores obtained from allele frequency data". Why?

Line 205: you present some mutations and describe some errors in prediction, such as G256E - “benign” and G256R - "pathogenic".

In this case, this can be explained by adding a positive charge generating pathogenicity, so a negative charge would be the opposite. Would your model have the ability to detect this? Of course, glycine is a special amino acid, so changing it to anything else can cause serious problems.

Line 214: the paper states that D488E is pathogenic (this is also discussed in line 402). This is strange, as replacing D with E doesn't impact the structure much (both are polar negative). Is there any discussion in the literature explaining why this mutation is considered pathogenic? Could you talk more about this?

Line 652 - describe all the supplementary materials.

Line 665 - what is the GitHub link?

Minors:

where you see (line 19):

 elusive in absence of clinical 

change to:

 elusive in the absence of clinical 

where you see (71):

 overcoming limitations of existing tools

change to:

 overcoming the limitations of existing tools

where you see (104):

 KCNQ2_Index [23], DeMag [38] and MAVERICK [39]

change to:

 KCNQ2_Index [23], DeMag [38], and MAVERICK [39]

where you see (140):

 In particular Logistic Regression, Support Vector Machine, Random Forest and Gradient Boosting algorithms 

change to:

 In particular, Logistic Regression, Support Vector Machine, Random Forest, and Gradient Boosting algorithms 

where you see (159):

 Higher values indicate larger average 

change to:

 Higher values indicate a larger average 

where you see (194): 

 the protein with unknown function

change to:

 the protein with an unknown function

where you see (206):

 G256W, G256V and G256R

change to:

 G256W, G256V, and G256R

where you see (213):

 Map of variants with highest ratio of 

change to:

 Map of variants with the highest ratio of 

where you see (227):

 We used the algorithms trained with the best performing version of the VFI score, in this case those with a standard 

change to:

 We used the algorithms trained with the best-performing version of the VFI score, in this case, those with a standard

where you see (247):

 a variant is identified as pathogenic it is 

change to:

 a variant is identified as pathogenic, it is 

where you see(259):

 red, dark purple and orange, respectively. 

change to:

 red, dark purple, and orange, respectively. 

where you see (495):

 (i.e. ~10% of potential missense variants)

change to:

 (i.e., ~10% of potential missense variants)

where you see (596):

LightGBM, CatBoost and XGBoost

change to:

LightGBM, CatBoost, and XGBoost

where you see (606):

 with highest performance was selected

change to:

 with the highest performance was selected

Author Response

Referee 2

In lines 85-88: you define "benign vs pathogenic" as labels, but in lines 213-216 (and 230): you start using "pathogenic vs tolerated". In line 252, severe or benign become subclasses of pathogenic.

Would it be possible to standardize the labels used? The current version is causing some confusion.

Thank you very much for highlighting this issue. We have amended figure legend 4, replacing “benign” with “tolerated”. In addition, we have added in page 2 the following sentence: “Although most databases classify variants as “pathogenic” or “benign”, we refer to these “benign” variants as “tolerated”, to emphasize the difference between variants which do not cause pathologies in patients, and pathological variants related to benign familial neonatal epilepsy or more severe manifestations.”

Table 2 indicates that they obtained 100% specificity in the test. However, in Table 3, you obtained high sensitivity and low specificity in the training stage using cross-validation, but the values were close in the test. I didn't understand the modification made to make this happen. Could you describe it better?

The values in Table 3 contain results for the phenotype prediction task. For this task, we observed that despite good AUC-ROC scores, the predictions made by the models were highly biased towards severe variants. To amend this, we modified the threshold until better balance was obtained, and this modification was then validated in the test set, which is why we observe better balance there. The values in Table 2 contain results from the pathogenicity prediction task, where it was not necessary to modify the default threshold of 0.5.

To clarify this issue, we have included in Table 3 legend, the following sentence: “Test set scores are reported after the threshold is modified to optimize the balanced accuracy of the model on the training set, which balances the sensitivity and specificity of the model.”

Line 143: you affirm that the "pLDDT score generated by AlphaFold2 is highly correlated with the VFI scores obtained from allele frequency data". Why?

This is a very intriguing observation; we do not have an explanation. It appears that this observation cannot be generalized. To make this clear, we have amended the last paragraph of page 6 as follows:

An interesting observation is that, in the particular case of KCNQ2, the pLDDT score generated by AlphaFold2 is highly correlated with the VFI scores obtained from allele frequency data, and when plotted against sequence positions, the pattern it generates is also similar to the VFI score, as can be seen in Figure 3A. However, VFI has higher resolution compared to pLDDT, meaning that it contains higher position dependent variance, while the overall pattern is very similar. Thus, we also tested whether replacing the VFI score with pLDDT would have a significant impact on the models performance by training another set of models using only pLDDT instead of VFI.

Line 205: you present some mutations and describe some errors in prediction, such as G256E - “benign” and G256R - "pathogenic".

In this case, this can be explained by adding a positive charge generating pathogenicity, so a negative charge would be the opposite. Would your model have the ability to detect this? Of course, glycine is a special amino acid, so changing it to anything else can cause serious problems.

The reason of failing to predict G256E as “tolerated” is unknown. This is a very particular situation that our and other models are not able to predict properly. This particular residue is located in the extracellular loop connecting transmembrane S5 with the selectivity filter, and is completely solvent exposed, not making contact with any other residues according to several Cryo-EM structures. Thus, we can speculate that a K at that position disrupts the linker in such a way that makes the pore inoperative, whereas an E does not. However, this is not captured by the features that our model and others are trained with. We believe that approaches such as all atom molecular dynamics simulations may help understand the basis for this observation, but this is beyond the scope of this manuscript.

 Line 214: the paper states that D488E is pathogenic (this is also discussed in line 402). This is strange, as replacing D with E doesn't impact the structure much (both are polar negative). Is there any discussion in the literature explaining why this mutation is considered pathogenic? Could you talk more about this?

We agree with the referee that the impact of this particular mutation is very intriguing. As requested, we have added more information regarding this mutation in the discussion, as follows:

For instance, none of the models, including the best-performing ones, accurately predicted the D488E mutation as pathogenic. A patient carrying this mutation experiencing seizures two months after birth, exhibited delayed development, and remained seizure-free for more than four years at the time of the report [49]. This conservative substitution, involving a negatively charged residue replaced by another negatively charged residue, is conserved across several fish species and removing a large stretch of amino acids containing this residue results in protein variants with similar electrophysiological properties as the wild type [50]. This residue is located in a region predicted by AlphaFold2 to adopt an alpha helical disposition, and flanked by A501, where the hard to predict pathogenic variant A501P maps. Interestingly, this mutation linked with severe encephalopathy has been found to be inherited from healthy parents [51]. The impact of these variants in health suggests that this putative helix performs an essential unknown role. Thus, some variants disrupt functions that are not yet fully understood.

Line 652 - describe all the supplementary materials.

Thank you very much for pointing out this omission. We have added a text file with the description of the supplementary material.

Line 665 - what is the GitHub link?

We have added the link before the references section:

Data Availability Statement: Data and code generated during this study will be made available on GitHub at https://github.com/BilbaoComputationalBiophysics/MLe_KCNQ2 at time of publication.

We have amended the list of typographical errors pinpointed by the referee. We would like to thank the referee for taking the time to compile this list. Thank you very much, indeed.